# Closed loop deep Bayesian inversion: Uncertainty driven acquisition for accelerated MRI

## Abstract

This work proposes a closed loop, uncertainty-driven adaptive sampling framework (CLUDAS) for accelerating magnetic resonance imaging (MRI) via deep Bayesian inversion. By closed loop, we mean that our samples adapt in real-time to the incoming data. To our knowledge, we demonstrate the first generative adversarial network (GAN) based framework for posterior estimation over a continuum sampling rates of an inverse problem. We use this estimator to drive the sampling for accelerated MRI. Our numerical evidence demonstrates that the variance estimate strongly correlates with the expected mean squared error (MSE) improvement for different acceleration rates even with few posterior samples. Moreover, the resulting masks bring improvements to the state-of-the-art fixed and active mask designing approaches across MSE, posterior variance and structural similarity metric on real undersampled MRI scans.

## 1 Introduction

Myriad of applications in control, data processing, and learning—from platform navigation to data mining and from channel estimation to compressive sensing (CS)—involve a linear projection of signals or data points into lower-dimensional space. In this dimensionality reduction, the measurement or the sensing matrix determines how much information we acquire per measurement, the ensuing computational ease of processing (since algorithms use the sensing matrix and its adjoint as subroutines), and the recovery guarantees.

In a resource constrained setting, these utilities create a Pareto trade-off wherein improving one worsens another. To impact all these fronts simultaneously, adaptive sensing (or sequential experimental design, active learning, etc.) aims to close the loop between the data acquisition and the inference, for instance, by exploiting information collected in past samples to adjust the future sampling process. While adaptive procedures promise great improvements over non-adaptive methods, they are too computationally demanding for real-time online response.

In the context of magnetic resonance imaging (MRI), the dimensionality reduction process (i.e., undersampling in the Fourier domain, often referred to as k-space) directly correlates with patient comfort, as it results in shorter scanning times. For the last decade, approaches motivated by compressed sensing have enabled successful reconstruction from highly accelerated (i.e., subsampled) data (see Lustig et al. (2007); Ravishankar & Bresler (2011b); Lingala & Jacob (2013); Otazo et al. (2015); Jin et al. (2016) and references therein). While compressed sensing prescribed fully random sampling (Candès et al., 2006; Donoho, 2006) for recovery, most CS-inspired approaches to MRI departed from this paradigm and relied on the heuristics of variable-density sampling (VDS) (Lustig et al., 2007).

There, the sampling pattern (or mask) is picked at random from a probability distribution that reasonably imitates the energy distribution in Fourier space, whereas fully random sampling of the Fourier space ignores this important structure in the signal, which leads to practically poor results (Lustig et al., 2008). VDS appears as a heuristic middle ground for a sampling pattern to incorporate the structure of energy distribution in Fourier space, while preserving the benefits of incoherent sampling. In VDS, the probability distribution considered has traditionally been parametric (Lustig

et al., 2007; Chauffert et al., 2014; Boyer et al., 2016) or constructed from data (Knoll et al., 2011; Vellagoundar & Machireddy, 2015; Bahadir et al., 2019).

The CS-inspired methods shift the burden from acquisition to reconstruction, as most of these methods are iterative, preventing online reconstruction of accelerated data. This slow reconstruction rendered the problem of optimizing Fourier space sampling prohibitively expensive, so little work was devoted to general method of designing sampling patterns for generic sampling methods (Gözcü et al., 2018).

In recent years, deep learning applied to MRI enabled high quality reconstruction for unprecedented acceleration rates (Wang et al., 2016; Schlemper et al., 2017; Hammernik et al., 2018), as well as near-online reconstruction times. This also led to a freshly renewed interest in the problem of optimizing the Fourier space sampling pattern (Bahadir et al., 2019; Weiss et al., 2019; Sherry et al., 2019; Jin et al., 2019; Zhang et al., 2019), although most of the research energy is still focused on developing more efficient reconstruction methods.

In its current stage, deep learning applied to MRI suffers from three main drawbacks: **(i)** as mentioned, most of the research energy has been devoted to more efficient reconstruction methods, despite recent results showing that the sampling masks used have a significant effect on the quality of reconstruction (Gözcü et al., 2018; Gözcü et al., 2019), **(ii)** assessing the reliability of the prediction of a reconstructed image is difficult for a clinician, due to the black-box nature of deep learning methods and **(iii)** the commonly used metrics for assessing the quality of the reconstruction (e.g. MSE, PSNR, SSIM) do not align with what clinicians see as valuable (Cheng et al., 2019).

A recent work of Adler & Öktem (2018) successfully demonstrated that a conditional Wasserstein GAN (cWGAN) (Arjovsky et al., 2017) can be used to learn the posterior distribution of images given undersampled measurements in a tractable fashion while only relying on samples from the joint distribution. This result provides a key opportunity to address all three drawbacks in the context of MRI by using the posterior variance of the reconstruction as an uncertainty estimate, which is a more natural criterion for image quality.

To this end, we show that a conditional WGAN can be trained on a continuum of inverse problems on various sampling rates, yielding an estimator of the posterior variance in the Fourier domain that can be used to drive the whole sampling process in a closed loop fashion. Despite the model being trained only for reconstruction and not sampling, the resulting variance estimator can reliably be used to guide a closed loop sampling procedure, thus providing patient-adapted sampling masks.

In particular, we demonstrate that the generated masks that minimize the uncertainty estimates in an online fashion reach similar reconstruction MSE as compared to the state-of-the-art fixed-mask approaches like Gözcü et al. (2018); Gözcü et al. (2019). While these approaches explicitly focus on minimizing MSE, we show that CLUDAS naturally outperforms them in terms of key visual metrics such as SSIM (Wang et al., 2004) without being trained on these metrics.

In addition, we investigate the reliability of our estimator in a wide range of undersampling regimes and show that even when using a few samples from the cWGAN posterior, the variance estimate is reliable enough to be used to drive the design of the mask. This makes it feasible to use **CLUDAS** in a closed loop adaptive setting, where our approach is competitive with an approach using an MSE-oracle on the testing set (which is not feasible on real problem without the ground truth available), and also beats strong open loop adaptive baselines while being easier to train and easier to apply.

**Contributions.**

- We show how to train a cWGAN to generate posterior samples across a continuum of sampling rates; We solve in a Bayesian fashion not only a single inverse problem, but a continuum of inverse problems.
- We demonstrate the first posterior-based mask design method for MRI.
- We propose to use the variance of the posterior distribution of images given measurements as quality metric for mask performance.
- We show that despite the network being trained as a *reconstruction* method and not as a sampling method, our adaptive approach CLUDAS is competitive in both settings and a strong contender for being used in active settings as it matches computationally expensive state-of-the-art approaches.

**Implications.** We contend that our uncertainty driven sampling framework can extend to many similar problems where one wants to reduce acquisition times, such as atomic force microscopy (Abramovitch et al., 2007), transmission electron microscopy (Kovarik et al., 2016), or trajectory optimization for ultrasound acquisitions (Malinen et al., 2005). Our results open the door of leveraging Bayesian experimental design methods for designing adaptive sampling patterns in clinical settings and beyond.

**Related works.** We especially want to highlight the work of Zhang et al. (2019), which bears several similarities with our method. It is important to note that their approach is *not* generative, as they only have point estimates of the mean and some learned uncertainty metric. Moreover, they assume the reconstructed image to be normally distributed with a diagonal covariance, a practically unrealistic assumption, which is not required in Adler & Öktem (2018).

## 2 NOTATION AND PROBLEM SETTING

**Notation.** Throughout this paper, we will refer to vectors as boldface lowercase letters $\mathbf{e}, \mathbf{x}, \mathbf{y}$, and assume that they correspond to $N \times N$ images that are vectorized to a $p$ dimensional space. In particular, we will assume that these vectors belong to some appropriately defined spaces $\mathcal{X}, \mathcal{Y} \subseteq \mathbb{C}^p$. We use vectors in $\mathbb{C}^p$ as MR images are inherently complex. Boldface uppercase letters $\mathbf{X}, \mathbf{Y}$ will in general refer to random variables (formally, they are random vectors), with the exception of $\mathbf{F}$ and $\mathbf{P}_\omega$ which will respectively refer to the discrete Fourier transform operator and the sampling operator, that will be defined below. Finally, we will use $\omega \in O = \{0,1\}^p$ to be a $p$ dimensional binary vector, and we will refer to it interchangeably as a *sampling, subsampling or undersampling pattern/mask*. Finally, we will refer to the distribution of a random variable $\mathbf{X}$ as $\mathcal{P}_\mathbf{X}$, and extend the nottation to conditioned random variables such as $\mathcal{P}_{\mathbf{X}|\mathbf{y}} \equiv \mathcal{P}_{\mathbf{X}|\mathbf{Y}=y}$.

**Problem setting.** An inverse problem is the task of recovering the ground truth $\mathbf{x} \in \mathcal{X}$ from measurements $\mathbf{y} \in \mathcal{Y}$. In the case of MRI, we consider the following acquisition model

$$\mathbf{y}_\omega = \mathbf{P}_\omega \mathbf{F} \mathbf{x} + \mathbf{e}, \tag{1}$$

where $\mathbf{y}_\omega \in \mathcal{Y}$ corresponds to the measurements obtained with a given sampling mask $\omega$ and where the sampling operator $(\mathbf{P}_\omega)_{ii} = 1$ if $i \in \omega$, $0$ otherwise. $\omega \subseteq [p] := \{1, \ldots, p\}$ is a set containing the sampled locations. Note that while the ground truth $\mathbf{x}$ is an *image* living in the *image space* $\mathcal{X}$, the measurements $\mathbf{y}_\omega$ live in the *Fourier space*. The Fourier space is often referred to as *k-space* in the MRI literature.

In this paper, we restrict ourselves to the setting where $\omega$ is composed of *lines* in the Fourier space, also known as Cartesian sampling in the MRI literature[1], and usually constrained by a maximal number $n$ of lines that can be acquired. $\mathbf{F}$ denotes the Fourier transform, $\mathbf{x} \in \mathcal{X}$ is the ground truth image and $\mathbf{e} \in \mathcal{Y}$ is a white additive noise. Without loss of generality, neglecting basic sampling effects, such as magnetic field inhomogeneity and spin relaxation, we assume $\mathbf{e} = 0$ in the sequel.

As we will be working in a Bayesian framework, we define a random variable $\mathbf{X} \sim \mathcal{P}_\mathbf{X}$ from which ground truth, complete measurements are generated, distributed according to the unknown prior $\mathcal{P}_\mathbf{X}$. From this we also define the distribution $\mathcal{P}_{\mathbf{Y}_\omega}$ as well as the joint distribution $\mathcal{P}_{\mathbf{X}, \mathbf{Y}_\omega}$ and the posterior $\mathcal{P}_{\mathbf{X}|\mathbf{Y}_\omega}$, where $\mathbf{Y}_\omega = \mathbf{P}_\omega \mathbf{F} \mathbf{X}$. The posterior is especially of interest as for a fixed $\mathbf{y}_\omega \in Y$, $\mathcal{P}_{\mathbf{X}|\mathbf{y}_\omega}$ (short for $\mathcal{P}_{\mathbf{X}|\mathbf{Y}_\omega = y_\omega}$) represents the probability distribution of ground truths that are likely to have generated the observed data $\mathbf{y}_\omega$ with a given mask $\omega$.

## 3 BACKGROUND

### 3.1 MASK DESIGN IN MRI

**Fixed masks.** The overwhelming majority of data-driven mask design approaches work in an open loop fashion: the sampling mask is built using training data and kept fixed at inference time. Even

---

[1]This kind of structured acquisition originates from physical considerations and has the benefit of being easily implementable in practice.

the VDS paradigm, that prescribes sampling a mask at random from a parametric distribution, uses a fixed mask that is tuned in an ad-hoc fashion when applied clinically (Jaspan et al., 2015).

Formally, we consider a training set of original data $\{\mathbf{x}_i\}_{i=1}^m$ that are assumed to originate from an unknown prior distribution $\mathcal{P}_{\mathbf{X}}$. We are constrained to a maximal sampling budget $n$ and want to find a mask that minimizes a given loss function $\ell$ (e.g. MSE, SSIM) on these training samples.

The abstract problem of our interest then can be written as follows

$$\min_{\omega \in \mathcal{A}} \mathbb{E}_{\mathbf{X} \sim \mathcal{P}_{\mathbf{X}}} \left[ \ell(\mathbf{X}, \mathbf{Y} = \mathbf{P}_\omega \mathbf{F} \mathbf{X}) \right]. \tag{2}$$

where $\mathcal{A}$ denotes the constrained set of masks $\omega$ that are made of lines (cf. section ) and that respect the maximal sampling budget, i.e. $|\omega| \leq n$ (here, $|\cdot|$ denotes the *cardinality* of the set $\omega$). Formally, let us define $\mathcal{S}$ as a set of subsets of $\{1, \ldots, p\}$ that contains the $N$ possible lines given a vectorized image of dimension $p = N^2$. $\mathcal{A}$ is defined as

$$\mathcal{A} := \{\omega \subseteq [p] : \omega = \textstyle\bigcup_{v \in S} v, S \subseteq \mathcal{S}, |\omega| \leq n\},$$

which means that $\omega$ is constructed as a union of the elements $v$ of a subset $S$ of all possible lines $\mathcal{S}$, with the additional constraint that the overall mask will respect the sampling budget $n$. However, the finite amount of samples requires to solve the empirical risk minimization version of Equation 2, namely $\min_{\omega \in \mathcal{A}} \frac{1}{m} \sum_{i=1}^m \ell(\mathbf{x}_i, \mathbf{y} = \mathbf{P}_\omega \mathbf{F} \mathbf{x}_i)$. Given that $m$ is large enough and that the training and testing data do originate from $\mathcal{P}_{\mathbf{X}}$, statistical learning theory guarantees that a mask performing well on the training set will adequately generalize.

In the literature, two trends are noticeable. A first body of works focused on constructing a good distribution from which to sample Ravishankar & Bresler (2011a); Vellagoundar & Machireddy (2015); Bahadir et al. (2019); Sherry et al. (2019). Other approaches tried to directly design a fixed sampling mask that performs well on training data (Seeger et al., 2010; Gözcü et al., 2018; Gözcü et al., 2019; Haldar & Kim, 2019). More recent approaches tried to jointly train the mask with a deep network Weiss et al. (2019); Bahadir et al. (2019).

**Adaptive masks.** Until the re-birth of deep learning, most reconstruction methods relying on CS suffered from long reconstruction times, due to their iterative nature. However, recent works leveraging the online reconstruction speed of deep learning achieved mask designs in a closed loop fashion, i.e., developing algorithms that could be used in an online fashion to adapt to patients.

For a fixed, unknown data $\mathbf{x}$, the adaptive approach aims at leveraging the information from the already measured frequencies to guide what should be acquired next. Instead of a fixed mask $\omega$, we are building up partial masks $\omega_t$ as union of individual Fourier space lines $v_t$, with $\omega = \omega_T$ being the largest mask satisfying $|\omega_T| \leq n$. The optimization now happens at runtime as a multistage problem which has to choose each new element $v_t$ such that

$$\min_{v_t | v_0, \ldots, v_{t-1}} \ell(\mathbf{x}, \mathbf{y}_{\omega_T} = \mathbf{P}_{\omega_T} \mathbf{F} \mathbf{x}; \ \omega_T = \textstyle\bigcup_{i=1}^T v_i), \tag{3}$$

that is, we want to take at each time step $t$ the sample that will allow us to get the lowest final error, but being constrained by our previous acquisitions and the fact that we cannot look into the future to know where we should sample. The problem requires developing an online sampling method that uses the partial information $\mathbf{y}_{\omega_t}$ at each $t$ to decide on its next action.

Two approaches have been proposed for this problem in the literature. Jin et al. (2019) took a self-supervised learning approach, where a sampling network learns to imitate a Monte-Carlo tree search method and predicts $v_t | v_0, \ldots, v_{t-1}$ for all $t$. Zhang et al. (2019) leveraged adversarial training to jointly train a reconstruction algorithm and an evaluator that gives scores to the quality of reconstructed lines in Fourier space. The sampling procedure simply iteratively added to the mask the lines with the lowest reconstruction score.

**Other types of sampling.** While our work here focuses on Cartesian sampling, which is by far the most widely used trajectory in MRI (Lustig et al., 2007), many other physically feasible trajectories have been investigated over the years for accelerated MRI. Radial trajectories have mainly been used in the context of dynamic MRI (Zhang et al., 2010a; Feng et al., 2014), and non-structured trajectories have also been explored and validated on real acquisitions (Lazarus et al., 2017). In particular, we note that some CS-based methods have shown online reconstruction times in the context of dynamic MRI (Zhang et al., 2010b) using radial trajectories, but such trajectories are rarely used in the context of static MRI.

# 4 METHODOLOGY

## 4.1 DEEP BAYESIAN INVERSION

In (Adler & Öktem, 2018), the authors propose a framework to estimate the posterior distribution $\mathcal{P}_{\mathbf{X}|\mathbf{Y}}$, i.e., the distribution of original images $\mathbf{x}$ that are likely to have generated the observed data $\mathbf{y}$. They formulate the problem as finding a parametrized generator $\mathcal{G}_{\theta^*} : \mathcal{Y} \to \mathbb{P}_{\mathcal{X}}$ that allows to minimize the Wasserstein distance with the unknown posterior $\mathcal{P}_{\mathbf{X}|\mathbf{Y}}$ over all the observations, namely minimizing

$$\min_{\theta \in \Theta} \mathbb{E}_{\mathbf{Y} \sim \mathcal{P}_{\mathbf{Y}}} \left[ \mathcal{W}(\mathcal{P}_{\mathbf{X}|\mathbf{Y}}, \mathcal{G}_{\theta}(\mathbf{Y})) \right]. \tag{4}$$

Here, $\mathbb{P}_{\mathcal{X}}$ is the space of probability measures on $\mathcal{X}$. As this approach in not tractable in practice, they show that Equation 4 can equivalently be formulated as

$$\min_{\theta \in \Theta} \left\{ \max_{\phi \in \Phi} \mathbb{E}_{\substack{(\mathbf{X}, \mathbf{Y}) \sim \mathcal{P}_{\mathbf{X}, \mathbf{Y}} \\ \mathbf{Z} \sim \eta}} \left[ D_{\phi}(\mathbf{X}, \mathbf{Y}) - D_{\phi}(G_{\theta}(\mathbf{Z}, \mathbf{Y}), \mathbf{Y}) \right] \right\} \tag{5}$$

After finding the optimal parameters $(\theta^*, \phi^*)$, the conditional generator $G(\mathbf{z}, \mathbf{y}) : \mathcal{X} \times \mathcal{Y} \to \mathcal{X}$ approximates the posterior distribution $\mathcal{P}_{\mathbf{X}|\mathbf{y}}$ and different values of $\mathbf{z}$ yield different samples from $\mathcal{P}_{\mathbf{X}|\mathbf{y}}$.

Note that Adler & Öktem (2018) applies this method to reconstruction data from ultra-low dose 3D helical computed tomography (CT). This differs from MRI in several regards, but it is significant in our case that we can generate different instances of Equation 4 by selecting different $\omega$ in Equation 1. Minimizing Equation 4 without giving $\omega$ explicitly would amount to generating a posterior distribution from data $\mathbf{y}$ acquired when using *any* possible mask $\omega$ sampled from a random vector $\Omega$ with a possibly unknown distribution. This is why the approach of (Adler & Öktem, 2018) minimizes over the distance for observation in Equation 4 and mask designs approaches consider minimization over original data (cf., equation 2).

**Uncertainty estimation.** Once the generator has been trained, we can sample from $G_{\theta^*}(\mathbf{Z}, \mathbf{y}_{\omega})$ which approximates $\mathcal{P}_{\mathbf{X}|\mathbf{y}_{\omega}}$. Let $\{\hat{\mathbf{x}}_i = G_{\theta^*}(\mathbf{z}_i, \mathbf{y}_{\omega})\}_{i=1}^{n_s}$ be samples of the posterior $\mathcal{P}_{\mathbf{X}|\mathbf{y}_{\omega}}$, where $\mathbf{z}_i$ are iid samples from $\mathbf{Z}$ and $n_s$ and the number of samples taken. Then, as in (Adler & Öktem, 2018), the ground truth image $\mathbf{x}$ can be estimated by the empirical point-wise mean of these samples, namely $\bar{\mathbf{x}} = \frac{1}{n_s} \sum_{i=1}^{n_s} \hat{\mathbf{x}}_i$. The corresponding empirical point-wise variance can be defined $\bar{\sigma}^2 = \frac{1}{n_s - 1} \sum_{i=1}^{n_s} |\hat{\mathbf{x}}_i - \bar{\mathbf{x}}|^2$, where $| \cdot |$ denotes the modulus.

Equally, one can estimate the ground truth Fourier spectrum $\mathbf{Fx}$ using the empirical average estimator $\mathbf{F}\bar{\mathbf{x}}$. The empirical point-wise variance in the Fourier space can be obtained as $\bar{\sigma}_{\mathbf{F}}^2 = \frac{1}{n_s - 1} \sum_{i=1}^{n_s} |\mathbf{F}\hat{\mathbf{x}}_i - \mathbf{F}\bar{\mathbf{x}}|^2$. This feature is specific to generative models, as getting samples from $\mathcal{P}_{\mathbf{X}|\mathbf{y}_{\omega}}$ allows to transform these to a different domain, enabling to have simultaneous variance estimates in both the image and Fourier spaces. This is not possible with methods that only provide point-wise estimates of the mean and the variance in image space, such as the one used by Zhang et al. (2019) or the direct estimation of Adler & Öktem (2018).

## 4.2 UNCERTAINTY DRIVEN SAMPLING

Due to the ability of constructing estimates of both the spatial and Fourier space pixel-wise variances of $\mathcal{P}_{\mathbf{X}|\mathbf{y}_{\omega}}$, the approach of (Adler & Öktem, 2018) can be leveraged to produce both fixed and adaptive sampling patterns by acquiring the frequencies with the highest empirical pixel-wise variance in the Fourier domain. As we constrained ourselves to acquiring full lines in the Fourier domain, we will consider the total estimated variance in the Fourier space along the $i$-th line $v_i \in \mathcal{S}$ and define

$$\mathbf{u}_i^{1D}(\mathbf{y}_{\omega}) = \sum_{j \in v_i} \bar{\sigma}_{\mathbf{F}, j}^2(\mathbf{y}_{\omega}). \tag{6}$$

Note that $\mathbf{u}_i^{1D}(\mathbf{y}_{\omega}) \in \mathbb{R}^N$ contains the variances of the $N$ possible lines in the Fourier domain. We will refer to $\mathbf{u}^{1D}(\mathbf{y}_{\omega})$ as the *aggregated* variance (along the $x$-dimension in Figure TODO).

**Fixed sampling.** Using the aggregated variance as a loss function, we can reformulate Equation 2 as

$$\min_{\omega \in \mathcal{A}} \mathbb{E}_{\mathbf{X} \sim \mathcal{P}_{\mathbf{X}}} \left[ \sum_i \mathbf{u}_i^{1D} (\mathbf{Y}_\omega = \mathbf{P}_\omega \mathbf{F} \mathbf{X}) \right] \tag{7}$$

where samples $\{\mathbf{x}_i\}_{s=1}^m$ are obtained from an unknown prior distribution $\mathcal{P}_{\mathbf{X}}$, and $\mathbf{Y}_\omega$ contains partial information on $\mathbf{X}$ through the model 1.

In practice, as $\mathcal{P}_{\mathbf{X}}$ is not available, one seeks to solve the empirical risk minimization (ERM) $\min_{\omega \in \mathcal{A}} \frac{1}{m} \sum_{s=1}^m \left[ \sum_i \mathbf{u}_i^{1D} (\mathbf{y}_{\omega,s} = \mathbf{P}_\omega \mathbf{F} \mathbf{x}_s) \right]$. The aggregated point-wise variance of the posterior can be seen as a cost that one seeks to minimize on a training set, and consequently, can it can replace traditional cost functions such as the $\ell_2$-norm in most fixed sampling optimization method.

**Adaptive sampling.** Ideally, we aim at making a series of sampling decisions $v_1, \ldots, v_T$ to minimize the total *final* uncertainty for a given ground truth image $\mathbf{x}$ once our sampling budget $n$ lines is spent, i.e., for each $t$,

$$\min_{v_t | v_0, \ldots, v_{t-1}} \sum_{i=1}^N \mathbf{u}_i^{1D} (\mathbf{y}_{\omega_T}), \tag{8}$$

where $\omega_T = \bigcup_{t=1}^T v_t$. Due to causality, we do not have access to this final posterior, or even the posterior of the mask which will result from choosing the next innovation $v_t$. We can only make use of the partial observations $y_{\omega_t}$ corresponding to the partial masks $\omega_t = \bigcup_{i=1}^{t-1} v_t$ up to the time step $t$. We choose to adopt a greedy approach to approximately solve

$$\min_{v_t \in \mathcal{S}} \sum_i \mathbf{u}_i^{1D} (\mathbf{y}_{\omega_t \cup v_t}) \text{ at each time step } t \tag{9}$$

by simply choosing as $v_t$ the line $i$ with the largest aggregated variance $\mathbf{u}_i^{1D}$. The overall flow is then: at each time $t$, (i) observe $\mathbf{y}_{\omega_t}$, (ii) select the line $v_t = v_{i^*}$, where $i^* = \operatorname{argmax}_i \mathbf{u}_i^{1D}(\mathbf{y}_{\omega_t})$, (iii) update $\omega_{t+1} = \omega_t \cup v_t$ and (iv) iterate until the cardinality constraint is met.

As no assumptions are made on the underlying distribution of the posterior, this application is only made possible by leveraging a generative framework that can estimate the posterior at all sampling rates considered for widely different mask designs. It is rendered tractable by the fact that even two samples from the posterior allow to construct an empirical variance estimate that can efficiently drive the sampling procedure.

## 5 IMPLEMENTATION

**Training data.** The data set used in the first three experiments (subsections) below consists of a proprietary dataset of 2D T1-weighted brain scans. In our experiments, we use 100 slices of sizes 256×256 from five such subjects, 20 per subject. Three subjects (60 slices) were used for training the network, two subjects (30 slices) for testing. The data were then massively augmented with both rigid transformations and elastic deformations to counter overfitting as our dataset is very small, following the recommendations of (Ronneberger et al., 2015; Schlemper et al., 2018). Exact details on the dataset and augmentation methods used can be found in Appendix A.1.

**Architecture** For posterior sampling, we used the same discriminator architecture as described in (Adler & Öktem, 2018). For the conditional generator, we used the cascading network of (Schlemper et al., 2018), where the data-consistency layer enforced perfect consistency. Perfect data consistency means that at the end of each block, one replaces the reconstructed value with the corresponding measured value in the Fourier space. This ensures that the reconstruction is consistent with the observations where measurements were acquired. We used 3 CNN blocks, where each block contained 5 convolutional layers followed by ReLu.

As our data are complex, we split the real and imaginary part as two channels and add two channels of Gaussian white noise to the conditional generator.

**Training.** We use the same loss as in (Adler & Öktem, 2018). The loss is reproduced in Appendix A.4 for completeness. We use Adam (Kingma & Ba, 2014) with $\beta_1 = 0.5, \beta_2 = 0.9$, and learning rate $2 \cdot 10^{-4}$ as in Adler & Öktem (2018) although we do not use noisy linear cosine decay

out of simplicity. The model is trained for $6 \cdot 10^5$ backpropagations, which was chosen adhoc to account for the fact there are combinatorial numbers of masks being observed for each image (our reference point Adler & Öktem (2018) uses $5 \cdot 10^4$ for a larger dataset).. For every 5 iterations, the generator was trained once and the discriminator was trained four times. In order to allow calculating the posterior throughout the sampling process, we generate observations of subsampled images at various rates by randomly generating horizontal Cartesian masks for sampling rates $\in [0.025, 0.5]$, as described in detail in Appendix A.2.

**Metrics.** We will use mean squared-error (MSE), structural similarity (SSIM) (Wang et al., 2004) as well as the posterior variance for comparisons. MSE and SSIM are computed between the reconstructed image and the corresponding original, ground truth image. The posterior variance is estimated through the a pixel-wise empirical variance estimate, and is averaged on the whole image to produce a single scalar. This metric does not require a reference.

## 6 EXPERIMENTS

Throughout our experiments, we use the empirical mean obtained from two posterior samples, as well as the corresponding empirical standard deviation. We show exhaustively in appendix B that while using 10 samples from the posterior improves the quality of reconstruction, it is sufficient to use the variance estimate from 2 posterior samples in the CLUDAS method.

### 6.1 CONSISTENCY OF THE UNCERTAINTY ESTIMATE

As can be Figure 1, the average MSE correlates well with the average posterior variance, both in image space and in Fourier space, which suggests that the posterior variance could serve as an approximate MSE oracle. While training was only performed in the image domain, the generated samples have consistency both in Fourier and in image space, showing that the uncertainty-based approach does provide meaningful information on the error in the reconstruction. The consistency in Fourier space is crucial for the sampling procedure, as our sampling method leverages the variance estimates in Fourier space, while the consistency in image space gives valuable information to interpret the reconstructed data, which are important for clinicians.

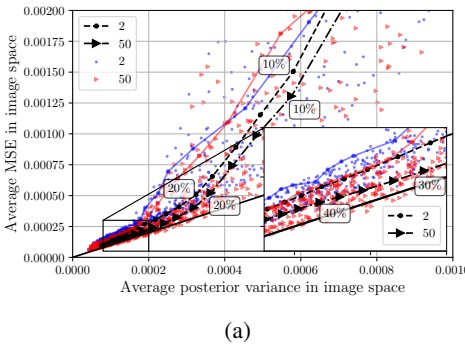
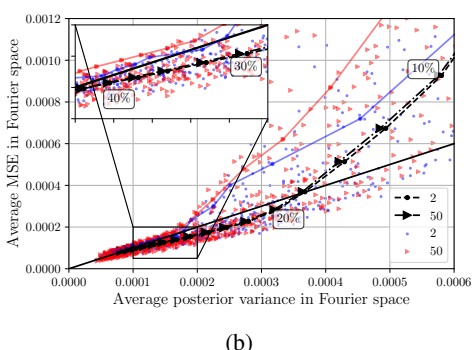

(a)                 (b)

Figure 1: Image (left) (resp. Fourier (right)) space MSE against image (resp. Fourier) space empirical posterior variance constructed with 2 and 50 posterior samples respectively. The coordinate of each point is given by the image (resp. Fourier) space MSE averaged over a reconstructed image and the pixel-wise image (resp. Fourier) space posterior variance averaged over this reconstructed image. The black lines represent the location obtained by averaging over a sampling rate over the whole testing set, with steps of $2.5\%$ sampling rate. The light red and blue lines are the example of a trajectory for a given image of the testing set when increasing the sampling rate.

### 6.2 RECONSTRUCTION QUALITY

In order to assess the reconstruction obtained by the adaptive masks, we define a closed loop oracle MSE driven adaptive sampling method (**CLOMDAS**), which leverages MSE instead of uncertainty at inference time. While CLOMDAS is not feasible in practice, it remains an interesting baseline showing how the mask design could be improved by having access to the actual MSE at testing time.

Figure 2 shows how the CLUDAS method compares against the CLOMDAS method on a sample from the testing set. CLUDAS is competitive with CLOMDAS at every sampling rate considered, even without having access to any oracle information. This behaviour is consistently observed on the whole testing set, as shown in Tables 1 below.

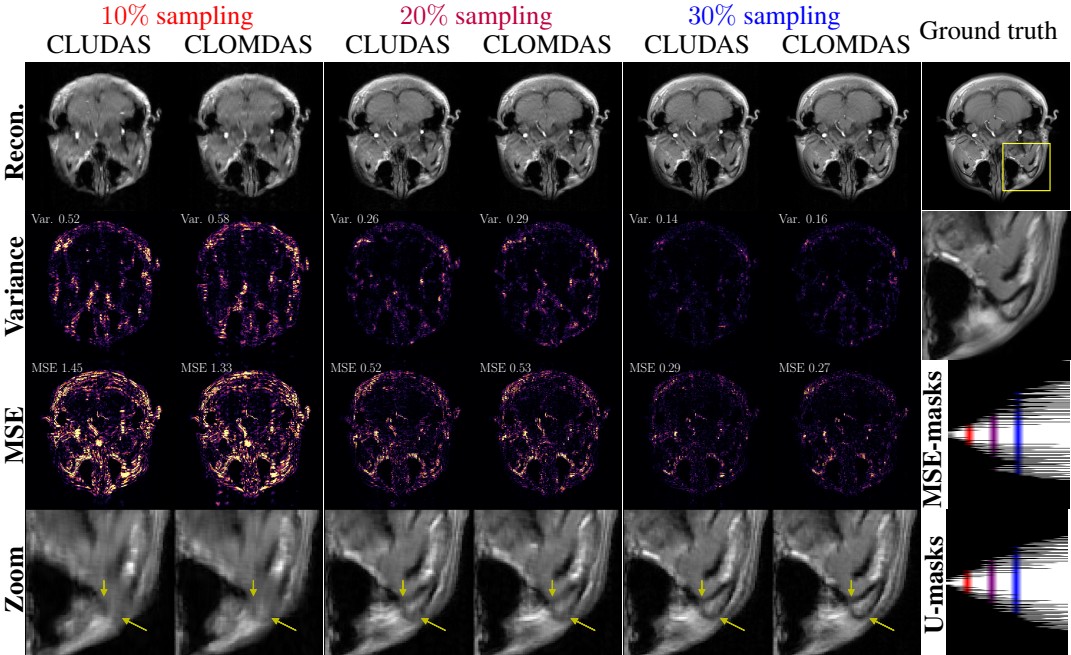

Figure 2: Comparison of reconstruction quality and variance estimation quality for CLUDAS (U-masks), as well as the CLOMDAS (MSE-masks), for different sampling rates. The zoomed-in data are taken at the location highlighted by the yellow square on the upper-right image. The MSE-/U-masks show the evolution of the masks with increasing sampling rate (x-axis). The data are averaged on 2 testing samples

## 6.3 COMPARISON WITH OTHER METHODS

We compare out method to the following

- **Learning based compressed sensing (LBC) (Gözcü et al., 2018; Sanchez et al., 2019):** This method incrementally builds up $\omega$ by computing an *expected* improvement of a loss $\ell$ at each step. This expected improvement is simply obtained by searching which line will add the largest improvement at the next step, and once it has been found, it is permanently added to the mask. Then, the algorithm proceeds until the cardinality constraint $|\omega| = n$ is met. When trained with MSE, we will refer to the method as LBC-M, and when trained to minimize variance, we will refer to it as LBC-V

- **Vellagoundar & Machireddy (2015):** This method proposed the simple heuristic approach of (i) constructing a PDF from a training data and (ii) sampling at random from the obtained PDF. Our implementation used the spectrum of the whole averaged training set for the PDF.

We were not able to compare our method to the closed loop method of Zhang et al. (2019), since their code is not being publicly available at the time of writing.

When comparing the performance of reconstruction of different mask designing methods on the modified generator of Adler and Öktem, we observe that the heuristic baseline of Vellagoundar & Machireddy (2015) performs significantly worse at any sampling rate and for any metric. This is not surprising, as this method simply samples art random from a constructed PDF. Comparing the variations of the LBC methods, we notice that increasing the number of averaged samples during the training phase of the mask is translated into a uniform improvment of the performance for any sampling rate. This is more exhaustively discussed in Appendix B . Focusing on the LBC methods

| Metric | MSE $\times 10^3$ | | | Variance $\times 10^3$ | | | SSIM | | |
|---|---|---|---|---|---|---|---|---|---|
| Sampling rate | 0.1 | 0.2 | 0.3 | 0.1 | 0.2 | 0.3 | 0.1 | 0.2 | 0.3 |
| Vellagoundar (2) | 3.73 | 1.34 | 0.47 | 1.12 | 0.59 | 0.30 | 0.67 | 0.80 | 0.87 |
| LBC-V(2) | 2.69 | 0.76 | 0.36 | 0.8 | 0.44 | 0.25 | 0.74 | 0.83 | 0.88 |
| LBC-V(10) | 1.68 | 0.55 | 0.32 | 0.65 | 0.36 | 0.22 | 0.77 | 0.86 | 0.89 |
| LBC-M(2) | 1.64 | 0.51 | 0.28 | 0.70 | 0.35 | 0.20 | 0.76 | 0.87 | 0.91 |
| LBC-M(10) | **1.43** | **0.50** | 0.28 | 0.63 | 0.34 | 0.20 | 0.79 | 0.87 | 0.91 |
| CLUDAS(2) *(Ours)* | 1.5 | 0.51 | **0.26** | **0.58** | **0.32** | **0.19** | **0.80** | **0.89** | **0.92** |
| CLOMDAS(2) | 1.39 | 0.49 | 0.26 | 0.61 | 0.34 | 0.21 | 0.81 | 0.90 | 0.93 |

Table 1: MSE scaled by $10^3$, posterior variance scaled by $10^3$ and SSIM on test data for different undersampling rates and mask design algorithms. The reconstruction is computed as the average of 2 samples. The lowest value (highetst for SSIM) for each undersampling rate is in blue. The value in parentheses corresponds to the number of samples used to compute the average.

trained with 10 posterior sample averaging, we see that the LBC-M method outperforms the LBC-V method. It is worth noting that the LBC-M uses the full ground truth to build its mask, while the LBC-V method only leverages the variance estimation in Fourier domain. This again highlight the reliability of the uncertainty as a mask designing technique. This conclusion is also supported by the CLUDAS approach remaining competitive with the CLOMDAS one, which is infeasible in practice, due to requiring oracle MSE calls at test time. Note also that our method does not require the heavy computational burden of generating the sampling mask ahead of time, and can immediately be used on-the-fly after training.

The CLUDAS method is the most effective at reducing posterior variance, even if LBC-M(10) and CLOMDAS(2) are close runner-ups. More surprisingly, our CLUDAS method is found to yield the best SSIM performance, a metric designed to match the human perception of quality better than MSE.

## 7 DISCUSSION AND FUTURE WORKS

**Posterior distribution for a continuum of sampling rates.** Successfully modelling the continuum of sampling rates stems from the fact the these inverse problems depend on each other in a highly structured and regular fashion. This enabled us to successfully demonstrate for the first time that a principled Bayesian approach for a closed loop mask optimization with rigorous variance estimates is feasible.

**Generically trained generative reconstruction method.** The current generative model was trained in a generic fashion and *not* specifically to optimize the quality of masks designed through it. The ability for designing masks stems purely from training it as a rigorous Bayesian modelling of the continuum

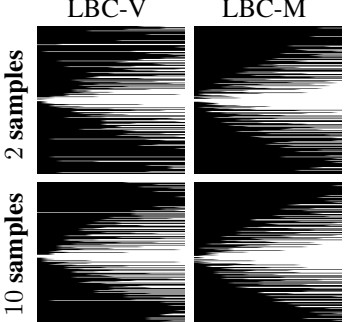

Figure 3: Comparison of fixed masks obtained by the learning-based method of Gözcü et al. (2018). The horizontal axis shows the mask growing as elements are iteratively added to it.

of inverse problems. This allows the posterior to be conditioned on incrementally collected information in a closed loop fashion. However, our method could easily be incorporated in a reinforcement learning-based framework aimed at jointly training reconstruction and sampling such as Jin et al. (2019). This would give the best of both worlds, giving principled uncertainty estimates to the RL sampler, moving beyond greedy sampling and possibly speeding up the training of the reconstruction method by focusing on regions with less reliable varaince estimates instead of using masks sampled from distributions as in this work.

**Limitations of the posterior estimation.** We leveraged the approach of Adler & Öktem (2018), which is the first of its kind to allow to construct a posterior estimator from samples of the joint distribution. While it works well empirically, the authors did not provide any analysis or guarantees on how well the generator captures the tails of the posterior distribution. Our observations suggest that unusual images, i.e. far away from the mean of the learned distribution might not be accurately

captured, i.e. the estimated variance is lower than expected. This could be due to the limited training data available, but might also an artifact of the cWGAN approach which tends to struggle with capturing weaker modes of their distributions. Specifically, the problematic examples might be an indication that while the loss shown in eq. (11) avoids mode collapse, there might still be some "mode deflation" leading to the network underestimating the diversity of the data distribution.

**Adaptive vs fixed sampling.** Adaptive and fixed sampling methods both have advantages and limitations from a practical perspective. The main advantages of a fixed mask approach lie in the ease of deployment of the obtained mask, as it simply needs to be programmed into a scanner. We also have a simple generalization bound of the obtained mask, relying on a simple application of Hoeffding's inequality. In contrast, adaptive methods are currently difficult to deploy it on scanners, as it would require hardware capable of running a neural network guiding the sampling procedure. They are also harder to train and currently lack rigorous reliability guarantees. However, if successfully trained and deployed they avoid rigid assumptions about the problem and are able to incorporate partial data into the acquisition process, which in turn leads to an improved performance. In this work, we found that using the adaptive method also increased robustness to the noise in the quality criterion used to drive the mask

Reliability guarantees beyond the variance estimate presented in this work (i.e. quantifying the uncertainty of the uncertainty estimate) are an important future direction of research.

**Extensions to the current model.** We showed that it is possible to use the uncertainty estimate to design fixed masks as in LBC-V, for settings where we are only interested in using the posterior variance as a natural criterion for reconstruction quality, e.g. masks for MRI systems where incorporating a neural network at scanning time is not feasible. In this setting, there are low hanging fruits for improving the method by making use of the available ground truth information, i.e. jointly using posterior variance with other metrics which require a ground truth.

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

## A    IMPLEMENTATION DETAILS

### A.1    TRAINING DATA

The data set used in all experiments consists of 2D T1-weighted brain scans of seven healthy subjects, which were scanned with a FLASH pulse sequence and a 12-channel receive-only head coil. In our experiments, we use 20 slices of sizes $256 \times 256$ from five such subjects, for a total of 100 slices. We use three subjects (60 slices) for training, two subjects (40 slices) for testing. Data from individual coils was processed via a complex linear combination, where coil sensitivities were estimated from an $8 \times 8$ central calibration region of Fourier space Bydder et al. (2002). The acquisition used a field of view (FOV) of $220 \times 220$ mm$^2$ and a resolution of $0.9 \times 0.7$ mm$^2$. The slice thickness was 4.0 mm. The imaging protocol comprised a flip angle of $70°$, a TR/TE of 250.0/2.46 ms, with a scan time of 2 minutes and 10 seconds. Following the recommendations of (Ronneberger et al., 2015; Schlemper et al., 2018), we then massively augmented the dataset to counter overfitting.

We apply both rigid transformations and elastic deformations. Specifically, at training time, each image was dynamically augmented with a randomly applied translation of $\pm 6$ pixels along $x$- and $y$-axes, rotations of $[0, 2\pi)$, reflection along the $x$-axis with $50\%$ probability. We also apply elastic deformations using the implementation in Simard et al. (2003) with $\alpha \in [0, 40]$ and $\sigma \in [5, 8]$.

### A.2    GENERATING OBSERVATIONS AT VARIOUS RATES OF SUBSAMPLING

We also dynamically generate horizontal Cartesian masks from sampling rates $\in [0.025, 0.5]$ by randomly selecting lines following suitably deformed Gaussians (see fig. 4).

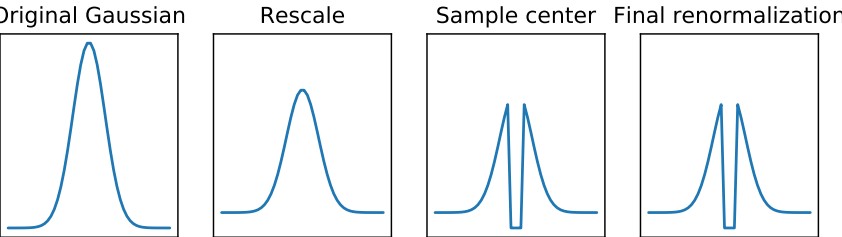

Figure 4:  Gaussian mask design used during training. We start with a plain Gaussian, rescale it to ensure selection in the tails,always sample the 4 highest energy lines in the center of the Fourier space, and finally renormalize and follow the modified Gaussian to sample the remaining lines.

This generation of training masks is biased towards the lower end of the frequency spectrum, and also does not consider extreme acceleration rates beyond 0.025. Ideally, one would use fully random subsampling across the whole range of subsampling rates, ensuring equally reliable variance estimates. In practice, extreme acceleration rates are unlikely to be used and most of the information is found in the lower frequencies, meaning these would almost surely be selected first. The present method represents a reasonable tradeoff between generalization across frequencies and training time.

### A.3    ARCHITECTURE

For posterior sampling, we used the same discriminator architecture as described in (Adler & Öktem, 2018). For the conditional generator, we used the cascading network of (Schlemper et al., 2018), where the data-consistency layer enforced perfect consistency. We used 3 CNN blocks, where each block contained 5 convolutional layers followed by ReLu.

As our data are complex, we split the real and imaginary part as two channels and add two channels of Gaussian white noise to the conditional generator. This also means that the discriminator of (Adler & Öktem, 2018) needs to be adapted to 6 input channels instead of 3.

## A.4 Loss function

The conditional WGAN must have its discriminator and generator trained alternatively, and Adler and Öktem proposed a novel discriminator loss that empirically avoids mode collapse. This discriminator takes three inputs instead of two in Equation 5, and is the one that we use in practice in the experiments. The general GAN loss reads

$$
\begin{aligned}
L_W(\theta, \phi) = \mathbb{E}_{\substack{(\mathbf{X},\mathbf{Y}) \sim \mathcal{P}_{\mathbf{X},\mathbf{Y}} \\ \mathbf{Z}_1, \mathbf{Z}_2 \sim \eta}} & \Big[ D_\phi\big(\mathrm{G}_\theta(\mathbf{Z}_1, \mathbf{Y}), \mathrm{G}_\theta(\mathbf{Z}_2, \mathbf{Y}), \mathrm{y}\big) \\
& - D_\phi\Big( \frac{1}{2}\big( D_\phi(G_\theta(\mathbf{Z}_1, \mathbf{Y}), \mathbf{X}, \mathbf{Y}) + D_\phi\big(\mathbf{X}, G_\theta(\mathbf{Z}_2, \mathbf{Y}), \mathbf{Y})\big)\Big) \Big]
\end{aligned}
\tag{10}
$$

Then, for a fixed $\theta$, the discriminator loss can be formulated as

$$
L_D(\phi) = L_W(\theta, \phi) + 10 L_{\mathrm{grad}}(\theta, \phi) + 10^{-3} L_{\mathrm{drift}}(\theta, \phi) + 10^{-4} \parallel \phi \parallel^2
\tag{11}
$$

where, $L_{\mathrm{grad}}$ is the gradient penalty term for the 1-Lipschitz constraint introduced in Gulrajani et al. (2017). The drift loss is used to stabilize training, to prevent the discriminator from being shifted to large values, as its performance is invariant to constant shifts. If the discriminator has a large constant, the overall loss (11) will not be influenced by the constant, so the drift loss penalizes large the expected squared norm of the discriminator.

For the generator, given a fixed $\phi$, the loss is defined as

$$
L_G(\theta) = -\mathbb{E}_{\substack{(\mathbf{X},\mathbf{Y}) \sim \mathcal{P}_{\mathbf{X},\mathbf{Y}} \\ \mathbf{Z}_1, \mathbf{Z}_2 \sim \eta}} [D_\phi\big(G_\theta(\mathbf{Z}_1, \mathbf{Y}), G_\theta(\mathbf{Z}_2, \mathbf{Y}), \mathbf{Y}\big)] + 10^{-4} \|\theta\|^2.
\tag{12}
$$

We refer the reader the Appendix D.3 of (Adler & Öktem, 2018) for the full discussion on the loss function.

## B   Greedy vs. adaptive

As can be seen in Figure 7, in the fixed mask (greedy) setting we require ten samples from the posterior before the uncertainty estimation yields competitive masks, while in the adaptive setting two samples suffice. The reason for this can be understood from considering the decision process and information flow of each algorithm, visualized in fig. 5. The adaptive algorithm uses only a single uncertainty estimate to make a decision, as it aggregates the estimate for each candidate $\mathbf{u}_i^{1\mathrm{D}}$ line in the Fourier space from a single pointwise uncertainty estimate. Due to the online nature of the sampling, if this estimate overshot or undershot, the algorithm will receive immediate feedback and can so revert to the mean, making it robust against a noisy estimator. In contrast to this, the greedy algorithm a) requires a separate uncertainty estimate for each candidate (since we want to look into the future and choose the candidate line which yielded the greatest improvement) and b) is performed on the training set only and then fixed. This means there is a good chance for a noisy estimator to spike in the uncertainty estimate and give the illusion of large improvements in uncertainty during the precomputation of the mask, and no way to compensate for mistakes at test time. This means we need a much more reliable estimator, increasing the number of posterior samples required.

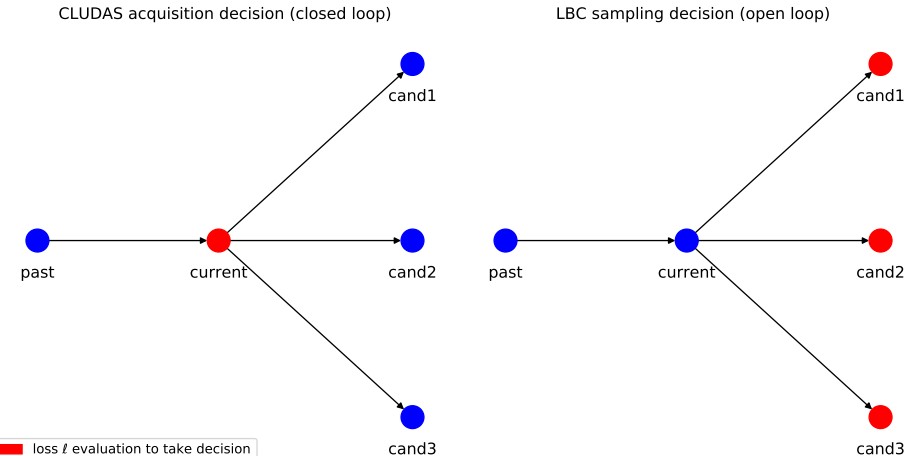

Figure 5: The adaptive sampling method (left) uses the loss heuristic $\ell$ (MSE, uncertainty) on *currently observed* information prior to the observation and then choosing based on this. In contrast to this, the greedy method wants to use the largest *expected* improvement in $\ell$ and so has to evaluate it in each possible future, effectively deciding a posteriori based on which yielded the actual greatest improvement.

## C  ADDITIONAL RESULTS

### C.1  EFFECT OF INCREASING THE NUMBER OF SAMPLES IN THE AVERAGE

Comparing adaptive baselines with 10 averaged samples in Figure 6, we see that the posterior variance magnitude is not affected by the increase in the number of samples, while MSE is simply shifted almost uniformly for each data. While the larger number of samples might affect how the variance is distributed on the image, it does not affect too much the variance obtained out of the sampling procedure.

Tables 2, 3 and 4 show the second experiment reproduced with averages computed with 10 posterior samples. The results are consistent with those of Section 6.3.

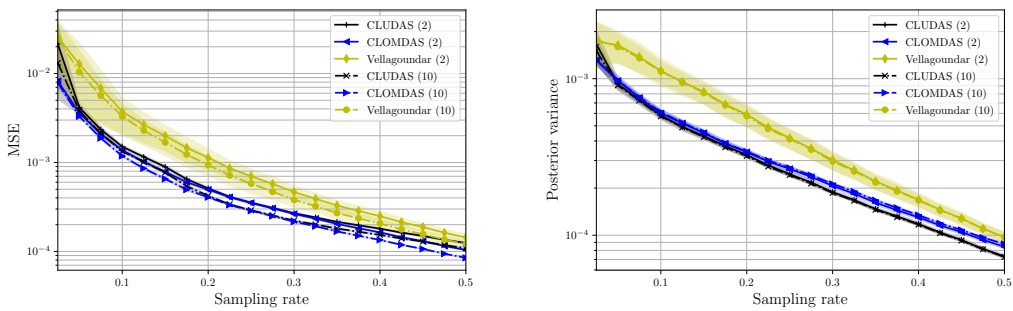

Figure 6: MSE (left) and image posterior variance (right) averaged over the whole testing set for different adaptive methods, average respectively on 2 and 10 samples from the posterior distribution.

### C.2  SUPPLEMENTARY RESULTS

Figure 7 shows the performance across all sampling rates of the methods considered in the main paper.

| Sampling rate | 0.05 | 0.1 | 0.15 | 0.2 | 0.25 | 0.3 |
|---|---|---|---|---|---|---|
| Vellagoundar (10) | 10.39 | 3.30 | 1.68 | 1.34 | 0.70 | 0.47 |
| LBC-V (2) | 8.98 | 3.48 | 1.63 | 1.00 | 0.58 | 0.34 |
| LBC-V (10) | 4.96 | 2.15 | 1.00 | 0.54 | 0.38 | 0.27 |
| LBC-M (2) | 4.26 | 1.45 | 0.76 | 0.43 | 0.3 | 0.23 |
| LBC-M (10) | 4.44 | **1.26** | **0.68** | **0.42** | **0.29** | 0.23 |
| CLUDAS (10) | **3.8** | 1.34 | 0.78 | 0.43 | **0.29** | **0.22** |
| CLUDAS (10) | 3.29 | 1.18 | 0.65 | 0.41 | 0.29 | 0.22 |

Table 2: MSE scaled by $10^3$ on test data for different undersampling rates and mask design algorithms. The reconstruction is computed as the average of 10 samples. The lowest MSE for each undersampling rate is in bold.

| Sampling rate | 0.05 | 0.1 | 0.15 | 0.2 | 0.25 | 0.3 |
|---|---|---|---|---|---|---|
| Vellagoundar (10) | 0.55 | 0.70 | 0.77 | 0.82 | 0.86 | 0.89 |
| LBC-V(2) | 0.55 | 0.7 | 0.78 | 0.82 | 0.86 | 0.89 |
| LBC-V(10) | 0.66 | 0.76 | 0.83 | 0.87 | 0.88 | 0.9 |
| LBC-M(2) | 0.66 | 0.79 | 0.84 | 0.89 | 0.91 | 0.92 |
| LBC-M(10) | 0.67 | 0.81 | 0.85 | 0.88 | 0.91 | 0.92 |
| CLUDAS(10) | **0.69** | **0.82** | **0.87** | **0.90** | **0.92** | **0.93** |
| CLOMDAS(10) | 0.71 | 0.83 | 0.88 | 0.91 | 0.93 | 0.94 |

Table 3: SSIM on test data for different undersampling rates and mask design algorithms. The reconstruction is computed as the average of 10 samples. The highest SSIM for each undersampling rate is in bold.

| Sampling rate | 0.05 | 0.1 | 0.15 | 0.2 | 0.25 | 0.3 |
|---|---|---|---|---|---|---|
| Vellagoundar (10) | 1.60 | 1.11 | 0.81 | 0.58 | 0.41 | 0.30 |
| LBC-V(2) | 1.47 | 0.95 | 0.68 | 0.54 | 0.40 | 0.27 |
| LBC-V(10) | 1.02 | 0.74 | 0.54 | 0.38 | 0.30 | 0.22 |
| LBC-M(2) | 1.20 | 0.70 | 0.51 | 0.35 | 0.25 | 0.20 |
| LBC-M(10) | 1.12 | 0.63 | 0.45 | 0.34 | 0.25 | 0.20 |
| CLUDAS(10) | **0.90** | **0.58** | **0.42** | **0.32** | **0.24** | **0.19** |
| CLOMDAS(10) | 0.98 | 0.62 | 0.46 | 0.34 | 0.27 | 0.21 |

Table 4: Estimated posterior variance scaled by $10^3$ on test data for different undersampling rates and mask design algorithms. The reconstruction is computed as the average of 10 samples. The lowest estimated variance for each undersampling rate is in bold.

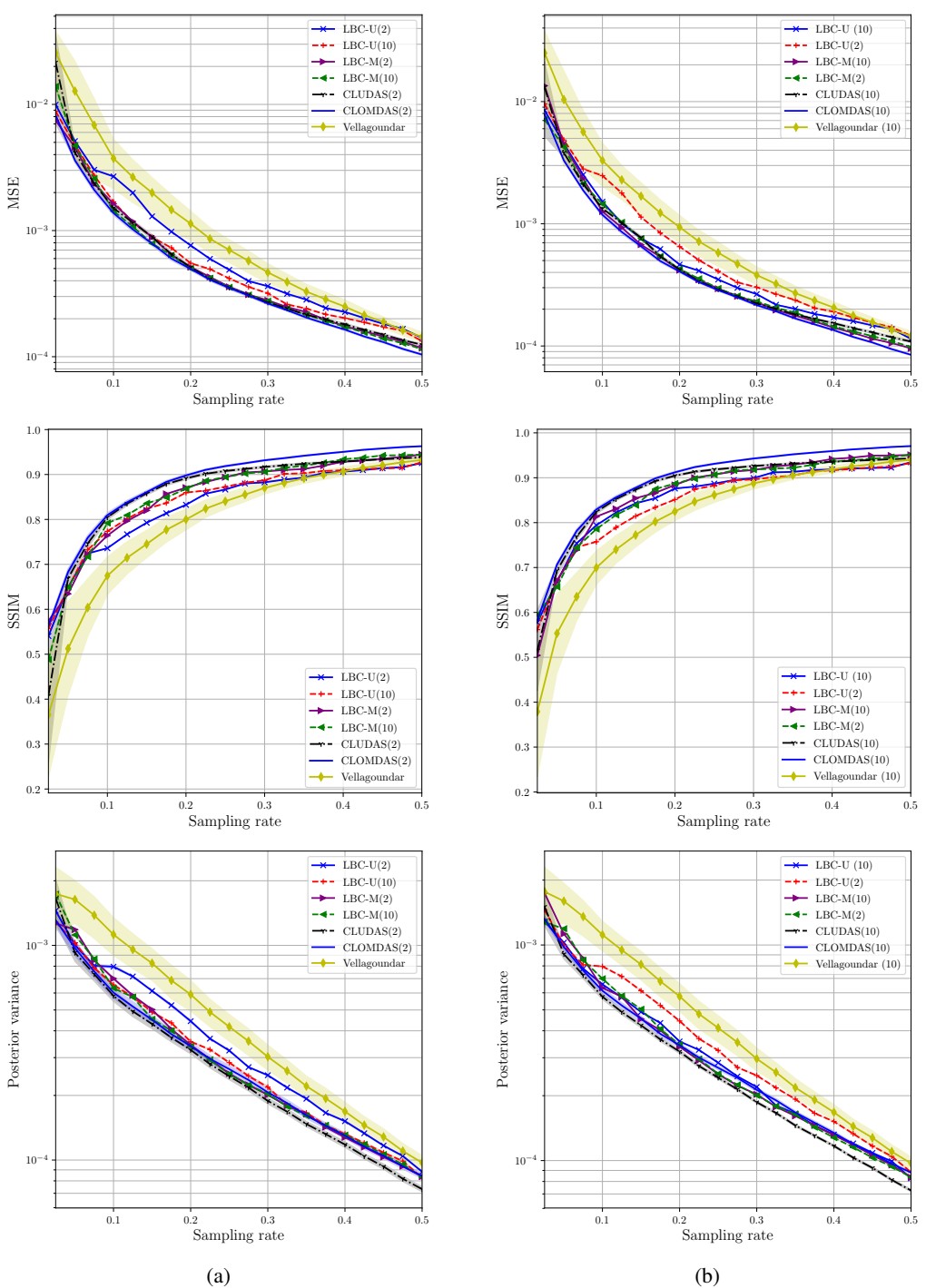

Figure 7: Performance of baselines for all sampling rates considered for MSE, SSIM and Posterior variance for (a) 2 averaged samples, (b) 10 averaged samples. The results are averaged across the whole testing set and several runs of each method was done to get error bars.

