# OpenReview forum: "Closed loop deep Bayesian inversion:  Uncertainty driven acquisition for fast MRI"
_ICLR.cc/2020/Conference — Reject_

### Official Review · AnonReviewer3 · 2019-10-19
**Official Blind Review #3**

**Rating:** 3

**Review:**

This paper proposes an active data acquisition framework for magnectic resonance imaging. A generative adversarial network is used to estimate the posterior distribution of the latent MRI image in a closed-loop and greedy manner where the uncertainty of the posterior image is used to guide the process.

The paper is well written and easy to follow. The main contribution seems to be the combination of deep Bayesian inversion in Adler & Oktem, 2018 with an uncertainty driven sampling framework. Using uncertainty to drive data acquisition and exploration is not a new idea; the concept has been applied to reinforcement learning, active learning, Bayesian optimisation, as instantiations of a broad class of methods in experimental design. The experimental results suggest that the technique can reduce the amount of time required to obtain good quality images from MRI scans which can potentially have a big financial impact. The technique is compared to several variants of compressed sensing approaches demonstrating superior performance.

My main concerns with the paper are:

1. The key idea of using uncertainty to guide sampling was also the main concept in Zhang et al. 2019. This submitted paper highlights differences in the models but does not provide an experimental comparison. Since both papers share the same concepts, this reviewer considers that a comparison is critical.

2. Deep Bayesian inversion approximates the posterior distribution by minimising the Wasserstein distance between the posterior  and a parametrised generator. I find the idea potentially powerful, with the advantage of learning a generative model as well, but wonder how this compares in theory and in practice to simpler stochastic variational inference and modern Hamiltonian MCMC. The min-max formulation is notoriously difficult to optmise and might lead to many local optima and instabilities.

3. Given the complexity of learning GANs and the sensitivity to initialization, results should contain more information such as the std of the MSE for several runs of the algorithm.


**Experience Assessment:**

I have read many papers in this area.

**Review Assessment: Checking Correctness Of Derivations And Theory:**

I assessed the sensibility of the derivations and theory.

**Review Assessment: Checking Correctness Of Experiments:**

I assessed the sensibility of the experiments.

**Review Assessment: Thoroughness In Paper Reading:**

I read the paper thoroughly.

---

> ### Author Response · Authors · 2019-11-12
> **Reconstruction and bayesian optimization/RL/active learning are different tasks.**
>
> Thank you for your comments and suggestions.
>
> *Summary: *
> - Reconstruction and bayesian optimization/RL/active learning are different tasks
> - For replication, generalization and dataset concerns, see general comments
> - Contribution remains the same, other generative approaches have not yet been shown and we expect them to struggle with the high dimensionality
>
> *Details:*
> Regarding the common on using uncertainty to drive optimization as in BO, there is a subtle distinction here. Our main contribution is a model which allows sampling from the posterior p(x|y), which is we use to perform reconstruction with pixel granular uncertainty quantification. This function can then be used to to drive adaptive sampling, but it is not a direct optimization task on any metric of reconstruction quality (SSIM, PSNR etc.) as would be required in order to apply BO, RL or active learning. Our CLOMDAS and CLUDAS algorithms represent just a simple approach that one would take in acquiring samples once one has access to a function which quantifies uncertainty of reconstructed regions (always sampling the most uncertain regions next), with CLUDAS being usable on real tasks (this is not the case of CLOMDAS). More elaborate versions built on this model could be imagined and in fact we are pursuing research in this direction.
>
> Regarding your first comment, we agree that a comparison with Zhang et al. would be critical, but their code is not available, even upon request. We will reuse the same data and reproduce as closely as possible their data processing pipeline, and we refer you to the general comment on the topic for more details.
>
> We see the suggestions from your second comment as valuable potential research directions, however the main contribution of our method remains the demonstration that using a generative approach is possible to model a closed-loop adaptive sampling mechanism. We expect that Hamiltonian MCMC will struggle with the high dimensionality , as MCMC are already prone to slow convergence.
>
> Regarding your final comment, we actually have provided error bars in the results of Figures 6 and 7 in the appendix (they are simply very small), but we will also add them to Table 1 for completeness.

---

### Official Review · AnonReviewer1 · 2019-10-22
**Official Blind Review #1**

**Rating:** 3

**Review:**

The paper proposes an uncertainty driven acquisition for MRI reconstruction. Contrary to most previous approaches (which try to get best reconstruction for a fixed sampling pattern) the method incorporates an adaptive, on-the-fly masking building (which is similar in spirit to Zhang at al. 2019). The measurements to acquire are selected based on variance/uncertainty estimates coming from a conditional GAN model. This is mostly an "application" paper that is evaluated on one dataset.

Strengths:
- The paper studies an interesting problem of adaptive MRI reconstruction
- The review of MRI reconstruction techniques is well scoped

Weaknesses:
- The evaluation is rather limited and performed on one, proprietary, relatively small sized dataset
- Some simple baselines might be missing


I like the idea of adaptive sampling in MRI. However, I'd slightly lean towards rejection of the paper. My main concerns are as follows:

The presentation of the paper could be improved. At the moment, the Theory section describes background information, related work and problem definition as well as the contribution of the paper. Maybe braking the section into related work, background and methodology (where the main contribution is presented) sections would improve the paper readability.

The paper uses a conditional GAN model (with a discriminator from Adler & Oktem, 2018 and a generator that is based on Schlemper et al. 2018). Making the methodological contribution to be rather limited.  The main difference w.r.t. the previous papers seem to be the last paragraph of  section 2.2 - the empirical variance estimation is performed in Fourier space.

A simple baseline to compare might be to train a Unet-like model (e. g. Schlemper et al 2018) with a Gaussian observation model (outputting a mean and a variance per each pixel) and train it to minimize Gaussian NLL. At the test time, one could simply sample from the Gaussian model instead of taking just the argmax of the output. It might be the case that the assumption of gaussian image might be too simplistic, however, it would be interesting to show it experimentally. Note that when sampling from such model the empirical variance estimation could be performed is the Fourier space too.

The experimental evaluation is rather limited and the dataset used in the experimental section is small. Adding another dataset would make the paper stronger.

Other comments:

There is a mention on training dataset and testing dataset -- there is no mention on validation set. How were the hyperparamenters of the conditional GAN selected?

As acknowledged by the authors, this paper bears several similarities with the work of Zhang at al. 2019. However, the approach is not compared to Zhang et al. Including this comparison would make the paper stronger.

It is interesting to see that CLUDAS outperforms CLOMDAS in terms of SSIM. If I understand this part properly, CLOMDAS uses ground truth image to estimate MSE. Is it expected that CLUDAS would outperform CLOMDAS?

Section 5, Adaptive vs. fixed mask: "We also have a simple generalization bound of the obtained mask, relaying on a simple application of Hoeffding's inequality." Could the authors add a citation or explain this part in more detail?


Some typos:
"...we aim make a series..."
".. define an closed-loop..."
"We choose adopt a greedy"
"... we we found that..."


**Experience Assessment:**

I have published one or two papers in this area.

**Review Assessment: Checking Correctness Of Derivations And Theory:**

I assessed the sensibility of the derivations and theory.

**Review Assessment: Checking Correctness Of Experiments:**

I assessed the sensibility of the experiments.

**Review Assessment: Thoroughness In Paper Reading:**

I read the paper at least twice and used my best judgement in assessing the paper.

---

> ### Author Response · Authors · 2019-11-12
> **Our contribution is about a closed-loop mask design system for MRI.**
>
> Thank you for your comments, corrections and suggestions.
>
> *Summary:*
> - Adler & Öktem have a fundamentally different task: sampling the full image space with low SNR vs. subsampling frequencies directly
> - MRI technical terms will be clarified/avoided when possible ; presentation will be updated and suggested experiments will be incorporated
>
> *Details:*
> We would like to highlight that our contribution lies mainly in the demonstration of a closed-loop system that can adaptively design sampling masks for MRI without requiring access to a ground truth (CLUDAS).  Adler and Öktem did not at all consider the problem of sampling mask design, as the problem of acceleration by choosing few locations where data are acquired is not present in CT. Rather, the challenge in CT is linked to obtain images with the lowest dose of radiation (i.e. data is sampled everywhere, but with low SNR). We also successfully demonstrate that the application of their method to MRI can be efficiently leveraged to provide an alternative to optimizing for MSE when designing sampling masks.
>
> The updated version that we will submit by the end of the week should contained a clarified presentation, and a more rigorous introduction of MRI-specific jargon. We will also try to incorporate your suggestion of using a Unet-like mode with a Gaussian observation model, and will also perform experiments on the DICOM dataset used by Zhang et al. However, as mentioned in the general comment, we cannot compare our results with Zhang et al. directly due to their code not being available, even upon request.

---

### Official Review · AnonReviewer2 · 2019-11-03
**Official Blind Review #2**

**Rating:** 3

**Review:**

The paper is quite well written and the idea is novel. However, the results are rather weak. The authors present a method to perform adaptive MR compressed sensing, i.e. decide online which readout to sample next. They compare it to an offline learning method where one sampling pattern is optimized for a whole training set, then applied to test data. The offline method performs better in terms of MSE, which is the loss it was trained for, meaning that the authors have not demonstrated a gain in adapting the sampling pattern to individual scans.

The primary concern with the paper is not with the author’s contribution, but with serious flaws in [Adler 2018] that unfortunately snowball into this one. While the authors in [Adler 2018] do acknowledge issues with learning a variance, they misdiagnose the problem as mode collapse. Mode collapse is an optimization problem, where the training set contains variability but the generator fails to learn it due to the lack of an encoder. That is not the case here: all the variability of the training set is encapsulated in y, and for each y the target empirical posterior distribution is a Dirac. This is very similar to the calibration problem in classification [1], where classifiers become overconfident because they are trained to always output 0 or 1. If the generator does not learn a Dirac, it can only be because of regularization (either explicit or implicit in the model architecture) or optimization failure (either involuntary or voluntary with early stopping.) Tweaking the loss as advocated in [Adler 2018] does not fundamentally change the problem as long as the loss is minimal at the target empirical posterior. It may change the dynamic behavior and result in posteriors with more variance when combined with early stopping, but those variances are not calibrated, i.e. they have not been trained to match the variances of the true continuous posteriors. In order to learn the variances, one would have to either provide multiple posterior samples for each y during training (not practical in this case,) or perform some kind of calibration on the validation dataset as in [1], i.e. learn the mean and variance from different data, which effectively uses the network’s interpolation properties as a proxy for true random sampling.

However this flaw does not invalidate the practical approach developed in the paper, but it seriously undermines its qualifications as a rigorous, principled, Bayesian approach. It also makes the reporting of posterior variances as final quality metrics pretty much useless since they are not interpretable: does lower variance mean that the generator got better at estimating the missing information, or that it got worse at estimating the true posterior variance? I would suggest to at least remove the variances highlights from Table 1 and Table 4, and maybe scrap the data altogether. The paragraph on posterior estimation should also be updated to represent the whole scope of the problem.

Section 2: Theory: suggest to remove “Without loss of generality”. Due to the known issues with variance estimation, having p(y | x) as a density instead of a Dirac could very well change the behavior of the generator.

Section 2.1 Adaptive masks. The whole first paragraph is somewhat misleading and should be revised. Real-time reconstruction is indeed possible without deep learning, see [2] for example. Furthermore, real-time reconstruction is not nearly fast enough for adaptive sampling. Real-time reconstruction means that reconstructing an image is at least as fast as scanning the whole image, i.e. in the order of 0.1 to 1s., but for adaptive sampling one must reconstruct at least as fast as the time between two successive readouts, i.e. in the order of 1 to 10 ms. Both [Jin 2019] and [Zhang 2019] only showed single-coil offline simulations with no indications of the reconstruction time and so do the authors.

Figure 1: I must be missing something here. How can the image-domain and Fourier-domain figures be different? The Fourier transform being orthogonal, norms and variances should be the same in both domains.

[1] C. Guo, G. Pleiss, Y. Sun and K. Q. Weinberger, “On Calibration of Modern Neural Networks”, ICML 2017 70:1321-1330.
[2] M. Uecker, S. Zhang, and J. Frahm, “Nonlinear Inverse Reconstruction for Real-Time MRI of the Human Heart Using Undersampled Radial FLASH”, MRM 63:1456-1462 (2010).


**Experience Assessment:**

I have published in this field for several years.

**Review Assessment: Checking Correctness Of Derivations And Theory:**

I assessed the sensibility of the derivations and theory.

**Review Assessment: Checking Correctness Of Experiments:**

I assessed the sensibility of the experiments.

**Review Assessment: Thoroughness In Paper Reading:**

I read the paper at least twice and used my best judgement in assessing the paper.

---

> ### Author Response · Authors · 2019-11-12
> **Some clarification is needed.**
>
> Thank you for your comments and suggestions.
>
> Summary:
> - We have some clarifying questions, GAN concerns and link to classifier calibration might not  be applicable?
> - Our model is a strong contender to be used in a real time fashion
>
> *Details:*
> Regarding the mentioned flaw in Adler’s paper, we are not sure to understand the argument of the reviewer, and we would like to ask you if you could make some points clearer - we note in passing that we are quite familiar with the GAN literature:
>
> - We are not sure what the reviewer refers to by “lack of an encoder” when mentioning that “Mode collapse is an optimization problem, where the training set contains variability but the generator fails to learn it due to the lack of an encoder.”
> - It is also unclear why “y the target empirical posterior distribution is a Dirac.” For us, the fact that we solve an inverse problem means a) we are interested in estimating x given y, not the inverse. The fact that we observe only a subset of fourier space means that there are inherently several possible x that could have generated the observed y on the manifold of data.
> - We are also unsure whether we understand the meaning of the statement: “Due to the known issues with variance estimation, having p(y | x) as a density instead of a Dirac could very well change the behavior of the generator.” Does the reviewer mean to criticize our noiseless setting, i.e. Y=FX+e, with e=0? If yes, we plan to address this by performing an experiment in which we add gaussian noise to the observation process.
> - As a final question, we do not quite understand the link the reviewer draws to classification calibration and uncertainty estimation via sampling from a posterior. Calibration refers to a classifier reporting meaningful confidence estimates when making predictions. We do *not* train to predict a specific variance, instead we sample from the posterior p(x|y) and obtain a distribution over images. We merely chose to *report*  a variance as a statistic which captures uncertainty. If the generator captures the posterior distribution well (measurable by reconstruction error and absence of mode collapse, then the quality of the variance estimate should follow. As stated in the general comment, we plan to add experiments to the appendix which show a) that no mode collapse occurs and b) the variance behaves as expected. We will also change the language to emphasize the fact we do not estimate an uncertainty or confidence directly.
>
>
> We appreciate your point on the term “real time”, but still think there is value in exploring methods which - if sped up sufficiently - can in principle support real time adaptive sampling. Our method requires roughly 8 ms to yield an estimation for a single data point averaged on two samples (which can be already reduced to 4ms by processing each sample in parallel). This is already close to the readout speed, in the order of milliseconds. Since our model does not rely on any ground truth or distributional assumptions to yield uncertainty estimates, and already requires roughly 4 ms to yield an estimation for a single data point, we consider it as a good adaptive sampling candidate, especially considering the increased industry focus on hardware acceleration of deep neural networks. We will add further timing data and a discussion of real time requirements to the appendix.

---

### Official Review · AnonReviewer4 · 2019-11-04
**Official Blind Review #4**

**Rating:** 3

**Review:**

The paper describes a method for accelerating MRI scans by proposing lines in k-space to acquire next. The proposals are based on posterior uncertainty estimates obtained from GAN-based reconstructions from parts of the k-space acquired thus far. The authors address an interesting and important problem of speeding up MRI scans and thus improving the subject's experience. The proposed method achieves better posterior uncertainty and SSIM scores than competing methods.



While the paper considers an important problem and takes a novel approach to solve it (using a GAN generative model to estimate uncertainty), I found that it may be particularly inaccessible to non-experts in the field of MRI image processing. Furthermore, several important methodology-related questions remain unanswered in the paper; and the experiments offered in the paper are insufficient for convincingly arguing the author’s claims.

Specifically, it is unclear whether GANs with their mode dropping behaviour are the right model choice for proposing reconstructions - they are likely to drop modes and by extension - yield overconfident uncertainty estimates. This can be expected to be particularly problematic for scans of images that differ from the training distribution, and is exacerbated by the fact that authors train the model only on scans from healthy subjects. Furthermore, because of the the GANs propensity to drop modes, it is also unclear whether the posterior variance numbers reported in the paper are directly comparable between the methods.

The method used for obtaining the uncertainty estimates from GAN samples implicitly makes the assumption that reconstructions follow a Guassian distribution with a diagonal covariance. This assumption is also made in a competing method of Zhang et. al (2019) that the authors do not compare against, and claim to improve upon methodologically (i.e. the authors state that the method of Zhang et. al (2019) cannot be used to produce uncertainty estimates in Fourier space). I am not convinced that the authors claims about the method differences are sufficiently substantiated (see more under major comments). And because the methods bear significant similarity to each other, an experimental comparison - which is currently missing - should be carried out (on open datasets that ideally include non-healthy subjects).

Finally, the paper teases fast(er) MRI in the title, but doesn’t touch on this topic in the text. This aspect should of the authors contribution be discussed at length, in particular comparing the Cartesian sampling strategy adopted by the authors to other strategies, as well as evaluating the feasibility of implementing the adaptive sampling strategy in an actual scanner (e.g. can the network be ran fast enough)?


==================
Major comments:
==================

1) In the proposed method the authors employ a procedure in which the currently sampled parts y of the k-space are fed to a generator network to obtain n_s reconstructions. These n_s sampled reconstructions are then averaged to obtain the empirical mean and variance, with the latter being used for estimating uncertainty. This procedure is potentially problematic for several reasons:

  a) First, taking the empirical mean and variance of the samples is in fact equivalent to assuming that the reconstructed image follows a Guassian distribution with a diagonal covariance. This is the same assumption the authors argue is not realistic when discussing the work of Zhang et al. (2019) in the end of Section 1.

  b) In case of GANs, which can model multi-modal distributions this uncertainty estimation is even more problematic in cases when the samples originate from different modes. What do the mean and variance represent then?

  c) As the authors highlight in the discussion section of the paper in the paper, GANs are prone to mode collapse. This is also potentially problematic for their estimator - in case of mode collapse their method would underestimate uncertainty. The fact that authors are able to use only two sampled reconstructions to estimate the mean and variance with acceptable accuracy is consistent with the occurrence of mode collapse in their generator. Furthermore, because mode collapse may occur in the author’s model, it is unsurprising that their method yields the smallest posterior variances in Table 1. The authors should provide evidence that mode collapse either does not occur or would not affect these numbers.

  d) Finally, the authors use the empirical mean and sampled reconstructions to obtain the empirical mean and variance in Fourier (k-) space. Specifically, they argue that “This feature is specific to generative models, as getting samples from P_{X|y_\omega} allows to transform these to a different domain [...] this is not possible with methods that only provide point-wise estimates of the mean and the variance in image space, such as the one used by Zhang et. al (2019)”. I don’t think this is true - Fourier transform is a linear transformation, thus given a mean and a variance in image space it is possible to deduce analytically what the mean and covariance in Fourier space would be. This should be elaborated in the text, and the comparison to Zhang et. al (2019) should thus be extended further.


2) The proposed method, CLUDAS, was evaluated against existing methods on a single proprietary dataset consisting of only 100 images from healthy individuals. This is potentially problematic, for several reasons:

  a) Using a proprietary dataset doesn’t allow follow-up works to compare against the author’s method; or for comparing CLUDAS to
 existing methods not considered in the paper; the methods should additionally be compared on a public dataset  and

  b) Applying the method only to a single (small) dataset does not allow for reasoning on how the method behaves in different data regimes. I strongly encourage the authors to apply their method on public datasets, for example on data used in Zhang et. al (2019) - this would then allow for comparing the two methods despite not having access to an implementation of Zhang et. al (2019).

  c) Since the data is used to train the GAN model is obtained from healthy individuals, it’s unclear whether it can be used to acquire data from subject that may potentially have aberrations in their scans - the GAN model would be expected to produce low uncertainty estimates for regions where these aberrations would lie and not propose acquiring parts of the k-space that could be used to resolve these aberrations. For similar reasons using a GAN that potentially drops modes (e.g. scans of unhealthy individuals, for example because they are less common in the training data) is also problematic. The authors should consider evaluating their method on a dataset that contains non-healthy subjects, and investigate the performance of the method when it’s evaluated on healthy subjects, but tested on unhealthy ones.

3) The title of the paper (“[...] for fast MRI”) suggests that the authors aim to accelerate the MRI data acquisition process.

  a) Yet they chose the work with the Carterian sampling (i.e. sampling lines in the k-space parallel to the x-axis), which arguably requires larger sections of the k-space to be sampled before a high-quality reconstruction can be obtained (e.g. see http://mriquestions.com/k-space-trajectories.html). Providing information on how this choice influences the speed of data acquisition (and thus subject’s comfort) is important in order to assess the applicability of the author’s method to real world scenarios. This information should be provided.

  b) The paper does not actually describe how MRI is sped up. Given that the acquisition budget (number of lines acquired in k-space) is fixed, and assuming that time per line is constant, it is unclear where the speed up comes in. More generally, the speed claims / aspect of the proposed method should be discussed in more detail - how is speed measured and evaluated? How does it compare to non-Cartesian sampling approaches.

  c) Finally, it’s unclear whether neural network inference can be made fast enough to allow for a real world application of the adaptive CLUDAS sampling - can the data be transferred fast enough from the scanner to do inference and propose the next line to scan without causing delays in the scanning process? This should be discussed in the paper.

4) Currently, the paper places a strong expectation of knowing about MRI and being familiar with MRI-specific terminology on the reader. This makes the work substantially less accessible to a wide audience with machine learning expertise as the common denominator. The authors should take steps towards making the text more accessible to non-(MRI) expert audience, for example by introducing some of the basic knowledge (e.g. the data acquisition and reconstruction processes in MRI) early on, departing from MRI-specific jargon (e.g. k-space, lines in k-space) in favour of ML terminology whenever possible, and taking care to define and possibly illustrate (strongly encouraged) the MRI-specific concepts (e.g. the k-space, lines in k-space, sampling masks). Some specific examples the authors should address follow.

  a) The k-space is defined only in passing as being the frequency domain. It’s unclear whether these lines (which correspond to sampling masks) in this domain are parallel to the x-axis. The math (e.g. Equation 1) and Figure 2 suggest that, but it’s not obvious.
x is referred to both as "model parameter" in Section 2 (and Equation 1) as well as the "ground truth image" later in the same section. This is confusing, especially because in case of GAN model parameter would typically refer parameters of the generator and discriminator.

  b) SSIM is not defined anywhere, but already used in the abstract.

  c) It could be made more clearly why undersampling is required in case of MRI. E.g. how/why does it correlation with patient comfort.

  d) “K-space sampling” is used already in the introduction, but not really defined.

  e) It’s unclear whether sampling, subsampling and undersampling all refer to the same concept or not.

  f) The term “sampling mask” is already used in the introduction, but not clear what it refers to.

  g) Use of the term “innovation” to refer v_t, which appears to be a one-hot vector marking the newly added line in k-space.

  h) The use of term “sampling decision” to refer to v_i.

  i) Unclear what eta in “Z ~ eta“ in Equation 5 refers to - it is not defined. Later in Section 2.2 it is stated that “z_i are independent samples form Z”. Is this the same as “z_i ~ eta”? If so, why the second layer of notation?

  k) In Introduction “[...] which is not feasible on a real problem without the ground truth available”. Unclear what the ground truth refers to, I assume it’s the ground truth image.

  l) In Section 2.1 data refers to x_i, with i=1,...,m; which I assume are images and thus contain real numbers. Yet Section 3 states “As our data are complex [...]”. This is confusing - is that different data?

  m) Sampling is used ambiguously in the paper - to refer to sampling in the k-space (e.g. sampling masks, sampling decisions v_1,...,v_n) and to refer to sampling reconstructions from the generator. This should be resolved to improve readability of the paper.

  n) Not entirely clear what k-space pixel-wise variances are. And what is the difference between spatial and pixels-wise variances is (Section 2.2).



==================
Minor comments:
==================

1. In the Introduction the authors argue that metrics such as MSE and SSIM “[..] do not align with what clinicians see as valuable.”, yet use these throughout the paper for evaluating and comparing methods. This decision should be explained.

2. In Section 2 (and beyond) images x are described as belonging to subspace C^p of complex numbers. While this is technically true, them to actually belong to the space of R^p. If so, this should be reflected in the text for the benefit of the readers. Furthermore, I don’t think dimensionality p is actually defined anywhere.

3. Compressive / compressed sensing abbreviations CS is defined twice in Section 1.

4. It’s not entirely clear why “The CS-inspired methods shift the burden from acquisition to reconstruction [...]”

5. Incorrect double quotes are used throughout the text (both are right quotes).

6.  In Introduction “[...] yielding an estimator which can be used to drive back the whole sampling process in a closed-loop fashion.”
it is unclear what it means to “drive back a sampling process”. Perhaps “back” should not be there?

7. In some cases it is unclear what the use of double quotes conveys, e.g.

  a) In Section 2.1 “[...] being the “full” mask [...]”

  b) In Section 2 “[...] ground truth “complete” images [...]”

  c) In Section 5 “[...] these inverse problems “depend” from each other [...]”

  d) Multiple places in Appendix B.

8. The convention of using small letters x, y for data samples / instances and capital letters X, Y for random variables could be made explicit.

9. Section mostly provides background (rather than theory) and could be named accordingly.

10. In Section 2.1, t refers to “time”. It may be clear if it were instead referred to as the step of the sampling process or something similar - the use of “time” to refer to some discrete set of actions can be a little confusing.

11. Section 2.1 refers to “[...] the online reconstruction speed of DL [...]”. This should be explained further - why are deep learning based approaches to reconstruction faster? Does this depend only on having the right hardware accelerators? Also, I don’t think the abbreviation “DL” was introduced.

12. Equation 4 is referred to both as an “Equation” and as a “Problem”.

13. In Equation 6 there is a summation over j from v_i. It is my understanding that v_i is a one-hot vector and it’s unclear what this summation means. Presumably it is the summation over pixels covered by line v_i, but the notation doesn’t convey this. It could be nice to also explain the “1D” superscript in this equation.

14. In Section 2.2 “[...] this is why the approach of (Adler & Oktem, 2018) minimizes over distance for observation in Equation 4 [...]”. I couldn’t follow the part about minimizing over the distance for an observation. Please consider making this more clear.

15. In Equation 8, what does index i run over?

16. Minor typos / textual issues:

  a) In Introduction “[...] of the our estimator [...]”

  b) In Introduction “[...] and show that even using a few samples [...]” -> even when using?

  c) In Section 2.2, after Equation 5 “[...] where t After finding the optimal”.

  d) Inconsistent use of “closed-loop” and “closed loop”.

  e) In multiple places throughout the paper a double space appears to be used instead of a single one.

  f) Section 4.1 “as can be in Figure 1”

  g) Section 4.3 “samples art random”

  h) In Section 5 “[...] “depend” from each other” should be depend on each other?

  i) In Section 5 “[...] we we found that [...]”

17. Unclear what’s meant by “[...] using the aggregated variance as a loss function” in Section 2.2

18. Section to refers to i* (integer scalar) as a line. Previously it was v_i (vector).

19. In Section 2.2 the authors state “Once the generator has been trained until convergence [...]”. The authors optimize a generator in an adversarial fashion. To my knowledge, this training procedure is not guaranteed to converge and would typically oscillate around a stable solution. Could the authors please comment on what they mean by convergence in this case and how they guarantee that the generators they train converge.

20. For the posterior variance results in Table 1 it should be discussed whether all the methods obtain / compute the variances the same way.

21. The “data consistency layer” (Section 3) should be explained briefly. How does it enforce perfect consistency and what’s meant by consistency here?

22. It should be made clear what MSE is calculated between in Figure 1 and Table 1. I assume it’s between ground truth (all frequencies sampled) and reconstructed (only some frequencies samples) images.

23. It’s not entirely clear what’s meant by consistency in Section 4.1

24. What do yellow arrows signify in Figure 2?

25. I don’t think the abbreviation UQ was defined in Figure 2.

26. The masks in Figure 2 are such that is a certain line was chosen at lower sampling rate (left on the x-axis), it would also be chosen at higher sampling rate (right on the x-axis). This is somewhat unexpected since the budget of lines v_i to be acquired differs between sampling rates. Why is there such consistency?

27. In Table 1, in the leftmost column, the number in brackets (number of posterior samples?) should be defined.

28. In Table 1 and Section 4.3: LBC-M and LBC-U - the -M and -U suffixes should be explained. What are the differences between the methods?

29. In Section 4.3: what are the FE methods?

30. Appendix A.3 is mostly a copy-paste from the main text. Unnecessary duplication?

31. Axes in Figure 4 (Appendix A.2) are not labeled or described.

32. Loss in A.4 (Equation) does not match Equation 5. In the former the discriminator takes three arguments instead of two, and the arguments z1 and z2 are not described.

**Experience Assessment:**

I do not know much about this area.

**Review Assessment: Checking Correctness Of Derivations And Theory:**

I assessed the sensibility of the derivations and theory.

**Review Assessment: Checking Correctness Of Experiments:**

I assessed the sensibility of the experiments.

**Review Assessment: Thoroughness In Paper Reading:**

I read the paper thoroughly.

---

> ### Author Response · Authors · 2019-11-12
> **Our approach does not make any Gaussian distribution assumption.**
>
> First of all, thank you very much for the exhaustive review and the many valuable comments.  We will answer each major comment separately.
>
> *Summary:*
> - Empirical means and variances via posterior samples do not place assumptions on posterior
> - We do not think our model has problems with mode collapse and will add experiments to verify this
> - Since we do not have place assumptions, analytic computations of Fourier mean and std isn’t possible analytically.
> - On replication, generalization and dataset concerns, see general comment.
> - Cartesian and radial sampling both have tradeoffs, we will clear this up together with improving definition of MRI technical terms
>
> *Details:*
> (1) For the first comment, we respectfully disagree that taking the empirical mean and variances implicitly assume a Gaussian distribution. The empirical mean and variance are simply statistics used to summarize the posterior distribution and do not place any distributional assumption on the posterior. This is a critical reason that distinguishes our contribution from the modelling of Zhang et al., where a Gaussian distribution is explicitly with diagonal covariance is assumed.
> Comments 1b) and 1c) are related to this: our mean and variance represent just this, a mean and a variance (i.e. variability) of a posterior distribution, which will represent all possible x which could have led to observation y.  If there are multiple modes (meaning source images) in the distribution, this is the model working as intended: it correctly captures the uncertainty and guides acquisition to its sources. Once there is enough information, we actually want the reconstruction variability to collapse to the ground truth, the image that should have generated the observed data.
> Of course mode collapse can still be problematic for our approach if it occurs in the sense that measurements of different ground truths get reconstructed as the same standard image, or if the reconstructions are always simpler versions without important diagnostic features, or the model becomes overconfident too quickly.  We do not think this is a problem in our model. WGANs tend to be more robust to mode collapse than all other types of GANS not designed specifically to avoid it (see e.g. the study in [1], which shows WGANs struggle with generation quality much more than with mode collapse). Both the standard reconstruction and  simpler version mode collapse would be detected with the error metrics (MSE, PSNR, SSIM….) and by visual inspection. We will perform further experiments to show that our model does not get confident too quickly and does indeed capture all modes of the distribution , see also the response to comment 2c) and the general comment.
> Finally, we respectfully disagree with 1d), as we have a general distribution at hand, we cannot analytically compute the transformation of its variance in Fourier. If it were Gaussian, we could have an analytical model of the dense covariance in the Fourier space and could sample from it directly. We will provide an experiment where we train our network to output a mean and variance and to minimize a Gaussian negative log-likelihood as suggested by reviewer 1.
>
> (2) Regarding the major comment 2), we will retrain our model on the publically available FastMRI dataset, with a procedure as close as possible to the one of Zhang et al. (cf. the general comment on this topic). We also refer you to the general comment on additional experiments for your comment 2c).
>
> (3) With respect to fast MRI, the title could be changed to accelerated MRI: the acceleration here originates from not acquiring entire lines in Fourier space. For Cartesian sampling, the acceleration obtained is linear: half of the lines not being acquired means that the scanning time is reduced by a factor 2. Cartesian sampling also has the advantage of being a fairly robust scanning trajectories, that is less susceptible to artefacts than for instance radial or spiral trajectories and consequently, they are the most widely used in the clinical setting; this is why we focused on it in this work. We will incorporate these explanations to the paper and will also discuss more in detail the inference time, adding a paragraph in our discussion on the subject.
>
> (4) Finally, we will try to thoroughly define the important MRI-terminology in a notations and definition section to clarify the presentation, and abstain from using the MRI-specific technical terms when possible.
>
> [1] Lala, Sayeri, et al. "Evaluation of mode collapse in generative adversarial networks." High Performance Extreme Computing, IEEE (2018).

---

### Author Response · Authors · 2019-11-12
**General comment**

First of all, thank you for the very detailed feedback provided.

We identified two main axes of improvement for our manuscript, which we believe we can address and include in the paper by the end of the week.

1. Improved clarity of presentation: We will reduce the MRI-specific language. We will delineate more clearly the theory section by breaking it into a) Notation and problem setting b) Background c) Methodology.

2. Additional experimental validation: We will provide experimental results on the fastMRI dataset [1], investigate potential mode collapse issues, clarify the training procedure, and provide a baseline in the like of a network trained to approximate a Gaussian negative log likelihood. The reviewers also unanimously asked for comparison with Zhang et al., which is not possible, due to their code not being publicly available - but we believe that the results on the fastMRI dataset should address these concerns. We also refer the reviewers to the additional comment on that matter.

[1] Zbontar, Jure, et al. "fastmri: An open dataset and benchmarks for accelerated mri." arXiv preprint arXiv:1811.08839 (2018).

---

> ### Author Response · Authors · 2019-11-12
> **Proper comparison with Zhang et al. (2019) not possible since their code is not available**
>
> Summary:
> - The code of Zhang et al. is not publically available, even upon request.
> - Their setup is also not very realistic since it uses magnitudes only, discarding all phase information
> - We will nonetheless try to replicate their setting as close as possible
> - We will also add baselines and an experiment that shows mode collapse and generalization are not an issue
>
> Details:
> The reviewers have pointed out the necessity of a comparison with the article of Zhang et al. (2019), with which our works bears similarities.
>
> We previously reached out to the authors, and sadly, their code is not publically available. However, Zhang et al. (2019) described that they used the DICOM files of fastMRI as the basis for their training and evaluation. We would like to draw the reviewers’ attention towards the facts
> 1. That the DICOM images used by Zhang et al. are only magnitude images, and discard all phase information. This introduces a Hermitian symmetry in Fourier space, and consequently, the sampling must be done symmetrically around the center in Fourier space (cf. section 2 in their supplementary material).
> 2. That the resizing of images that they performed also changes the distribution of Fourier space in an unpredictable fashion.
>
>
> **We will nonetheless retrain our model on this larger scale dataset following the methodology proposed in Zhang et al. (2019), resizing the images to 128x128, selecting the close-to-central images from each volume and normalizing the image with respect to the whole volume. We will however keep working with the complex data. **
>
> These two modelling steps taken by Zhang et al. introduces additional unrealistic assumptions, which could limit the applicability of their method in real life applications. In addition to this, as discussed in the paper, their assumption of a Gaussian distribution with diagonal covariance is not a realistic assumption.

---

### Decision · Program_Chairs · 2019-12-19

**Decision:**

Reject

**Comment:**

The author responses and notes to the AC are acknowledged.  A fourth review was requested because this seemed like a tricky paper to review, given both the technical contribution and the application area.  Overall, the reviewers were all in agreement in terms of score that the paper was just below borderline for acceptance.  They found that the methodology seemed sensible and the application potentially impactful.  However, a common thread was that the paper was hard to follow for non-experts on MRI and the reviewers weren't entirely convinced by the experiments (asking for additional experiments and comparison to Zhang et al.).  The authors comment on the challenge of implementing Zhang is acknowledged and it's unfortunate that cluster issues prevented additional experimental results.  While ICLR certainly accepts application papers and particularly ones with interesting technical contribution in machine learning, given that the reviewers  struggled to follow the paper through the application specific language it does seem like this isn't the right venue for the paper as written.  Thus the recommendation is to reject.  Perhaps a more application specific venue would be a better fit for this work.  Otherwise, making the paper more accessible to the ML audience and providing experiments to justify the methodology beyond the application would make the paper much stronger.